# Acceptability of a Patient Portal (Opal) in HIV Clinical Care: A Feasibility Study

**DOI:** 10.3390/jpm11020134

**Published:** 2021-02-16

**Authors:** Dominic Chu, Tibor Schuster, David Lessard, Kedar Mate, Kim Engler, Yuanchao Ma, Ayoub Abulkhir, Anish Arora, Stephanie Long, Alexandra de Pokomandy, Karine Lacombe, Hayette Rougier, Joseph Cox, Nadine Kronfli, Tarek Hijal, John Kildea, Jean-Pierre Routy, Jamil Asselah, Bertrand Lebouché

**Affiliations:** 1Department of Family Medicine, McGill University, Montreal, QC H3S 1Z1, Canada; dominic.chu2@mail.mcgill.ca (D.C.); tibor.schuster@mcgill.ca (T.S.); ab.abulkhir@gmail.com (A.A.); anish.arora@mail.mcgill.ca (A.A.); stephanie.long@mail.mcgill.ca (S.L.); alexandra.depokomandy@mcgill.ca (A.d.P.); nadine.kronfli@mcgill.ca (N.K.); 2Canadian Institutes of Health Research Strategy for Patient-Oriented Research Mentorship Chair in Innovative Clinical Trials in HIV, Montreal, QC K1A 0W9, Canada; david.lessard2@mail.mcgill.ca (D.L.); kedar.mate@mail.mcgill.ca (K.M.); kimcengler@gmail.com (K.E.); yuanchao.ma@muhc.mcgill.ca (Y.M.); joseph.cox@mcgill.ca (J.C.); 3Centre for Outcomes Research and Evaluation, Research Institute of the McGill University Health Centre, Montreal, QC H4A 3S9, Canada; 4Chronic and Viral Illness Service, Division of Infectious Disease, McGill University Health Centre, Montreal, QC H4A 3J1, Canada; jean-pierre.routy@mcgill.ca; 5AP-HP, Hôpital Saint-Antoine, Service des Maladies Infectieuses et Tropicales, 75012 Paris, France; karine.lacombe2@aphp.fr (K.L.); hayette.rougier.sat@aphp.fr (H.R.); 6IMEA, Institut de Médecine et d’Epidémiologie Appliquée, F75018 Paris, France; 7Department of Radiation Oncology, Cedars Cancer Centre, McGill University Health Centre, Montreal, QC H4A 3J1, Canada; tarekhijal@gmail.com (T.H.); jamil.asselah@mcgill.ca (J.A.); 8Medical Physics Unit, Gerald Bronfman Department of Oncology, McGill University, Montreal, QC H4A 3T2, Canada; john.kildea@mcgill.ca

**Keywords:** HIV, patient portal, patient-reported outcome measures, e-health, personalized medicine

## Abstract

Opal (opalmedapps.com), a patient portal in use at the Cedars Cancer Centre of the McGill University Health Centre (MUHC) (Montreal, Canada), gives cancer patients access to their medical records, collects information on patient-reported outcome measures (PROMs), and has demonstrated patient satisfaction with care. This feasibility study aims to evaluate Opal’s potential acceptability in the context of HIV care. People living with HIV (PLWH) and their healthcare providers (HCPs) completed cross-sectional surveys from August 2019 to February 2020 at large HIV centers, including the Chronic Viral Illness Service of the MUHC, and other HIV clinical sites in Montreal and Paris, France. This study comprised 114 PLWH (mean age 48 years old, SD = 12.4), including 74% men, 24% women, and 2% transgender or other; and 31 HCPs (mean age 46.5 years old, SD = 11.4), including 32% men, 65% women, and 3% other. Ownership of smartphones and tablets was high (93% PLWH, 96% HCPs), and participants were willing to use Opal (74% PLWH, 68% HCPs). Participants were interested in most Opal functions and PROMs, particularly PROMs capturing quality of life (89% PLWH, 77% HCPs), experience of healthcare (86% PLWH, 97% HCPs), and HIV self-management (92% PLWH, 97% HCPs). This study suggests Opal has high acceptability and potential usefulness as perceived by PLWH and HCPs.

## 1. Introduction

Human immunodeficiency virus (HIV), like other chronic conditions, requires consistent, long-term self-management by people living with HIV (PLWH), including engagement in care and adherence to antiretroviral therapy (ART) [1]. Increasing age, co-morbidities, and disabilities can increase the burden on PLWH and their multidisciplinary healthcare providers (HCPs) [2,3]. Additionally, PLWH may have experienced diverse psychosocial issues such as depression (34–42% of PLWH on ART), anxiety (21–40%), stigma or discrimination (42–83%), as well as unemployment, and limited formal education [2,4]. In turn, these factors can negatively affect access to and engagement in care as well as ART uptake [2,3,5]. 

To better support PLWH and their HCPs in the management of HIV, a promising solution is a patient portal, which is an extension of the electronic medical record system that provides patients secure access to their lab results, progress notes, and appointment schedules [6]. Patient portals can also include features to enhance communication with HCPs (e.g., text messaging), facilitate treatment access (e.g., medication refill request processing), and provide appointment or medication reminders. The functionalities and services of patient portals are relevant for clinical practice, as they have been reported to empower patients, improve engagement in care, and allow patients to make shared informed decisions with their HCPs [6,7,8,9,10,11,12,13,14,15,16,17]. Moreover, some patient portals allow for convenient electronic administration of patient-reported outcome measures (PROMs) [11,18], which provide health information from the patients’ perspective without revision or interpretation by a clinician [19]. PROMs can improve the clinical management of symptoms, side-effects, adherence, and psychosocial needs, among others [20,21], and are thus relevant in the context of HIV chronic care [2,10,22]. 

Opal (opalmedapps.com), an innovative and award-winning person-centered portal, was first piloted in 2018 at the Cedars Cancer Centre of the McGill University Health Centre (MUHC) [6]. The patient-facing component of Opal is a smartphone application that offers patients access to their personal health information (including clinical notes and laboratory test results) and appointment schedules. Opal also offers additional functions that promote self-management, including personalized educational material tailored to diagnosis and stage of treatment, and administration of PROMs. Opal is unique given that it was designed through a participatory stakeholder co-design approach; patients and HCPs were engaged in all stages of Opal’s development. In fact, it was a breast cancer patient and a McGill University computer science professor, the late Laurie Hendren, who identified the needs of patients that spurred the development of Opal [23]. Our goal is to adapt Opal to HIV care, as there is currently no HIV-specific patient portal in Canada. 

Opal was designed with oncology patients and thus may not be directly transferable to other health conditions, such as HIV. Oncology and HIV care are two medical specialties that differ greatly in terms of affected populations, treatments, and care providers involved. A lack of stakeholder involvement was a central reason for the failure of other early patient portals [12,24,25]. To ensure Opal’s uptake in HIV care [13], consistent with the approach initiated in oncology, HIV-specific stakeholder input was essential before offering Opal to PLWH [26]. The research question for this feasibility study was “How acceptable is the Opal patient portal for users in HIV care (i.e., PLWH and HCPs)?” This study also aimed to assess: 1) the experience of PLWH and HCPs with healthcare applications and smart device ownership, 2) PLWH and HCPs’ interest in Opal and their preferences for sharing personal health information, 3) the anticipated benefits and inconveniences of Opal, and 4) PLWH and HCPs’ interest in different Opal PROMs.

## 2. Materials and Methods

### 2.1. Study Design

This feasibility study employed a cross-sectional design. Feasibility studies that do not pilot aspects of an intervention or study methodology, as is the case here, attempt to answer questions about whether some aspect of a future trial is achievable [26]. This can include determining the acceptability of an intervention or the perceived importance of types of outcomes [26], which were among our objectives. Acceptability can be considered as the agreeable or interested views of stakeholders towards a specified innovation, such as Opal and its functions [27]. 

This study was conducted as part of a broader research program (the I-Score program) with sites in Canada and France aimed at improving ART adherence among PLWH using electronically administered PROMs [28]. In the HIV context, Opal will first be implemented in a pilot study at the Chronic Viral Illness Service (CVIS), one of the largest public hospital-based HIV clinics in Quebec, Canada, which provides comprehensive multidisciplinary care to over 1600 PLWH. Subsequently, our goal is to implement Opal across Montreal, Quebec, and in France as well; therefore, we have recruited participants from other Montreal-based HIV clinics and Hôpital Saint-Antoine (Paris, France). 

### 2.2. Study Sites

Recruitment focused on the CVIS. However, Opal is expected to be eventually implemented in other urban HIV care centers in Montreal and Paris; therefore, some PLWH and HCPs were recruited from Service de Maladies Infectieuses et Tropicales (SMIT) at Hôpital Saint-Antoine. HCPs were also recruited from Montreal-based non-CVIS sites specializing in HIV care, including the Centre Hospitalier de l’Université de Montréal, Clinique Médicale du Quartier Latin, and Clinique Médicale l’Actuel. Research ethics approval was obtained from the MUHC Research Ethics Board (study number: 2020-5910). Approval was obtained from the research ethics board of the Research Institute of the McGill University Health Centre, in Montreal, Canada, where two co-investigators are based. According to French public health legislation [29], no ethical approval was needed in France. A confidentiality and data transfer agreement was signed between l’Assistance Publique-Hôpitaux de Paris (AP-HP) at the Hôpital Saint-Antoine and the MUHC; thus, a separate REB for Hôpital Saint-Antoine was not required. Our study also meets the standards set by the Declaration of Helsinki.

### 2.3. Study Design and Participant Eligibility

Convenience sampling was used to recruit PLWH. To be included in the study, PLWH had to be over 18 years of age and receiving care for HIV, with no self-reported cognitive impairments. PLWH were recruited by referral from their HCPs during regular clinical appointments. The principal investigator recruited HCPs with at least 6 months of clinical experience in HIV care with purposeful sampling through personal invitation. HCPs included individuals who would be expected to use Opal to facilitate HIV care, comprising physicians, pharmacists, nurses, social workers, and administrative staff. 

### 2.4. Data Collection

Data were collected from August 2019 to February 2020. Two distinct surveys for PLWH and HCPs were developed to address each objective using validated tools from the literature [6,27,30,31,32,33]. 

The PLWH survey (73 items) (Appendix A) captured demographics and addressed our first objective, by measuring smart device use and healthcare technology self-efficacy [33]; our second objective, by evaluating interest in Opal’s functions and preferences for sharing personal health information [6,27,32]; our third objective, by collecting data on the anticipated impact of accessing physicians’ clinical notes [31]; and our fourth objective, by acquiring PLWH’s interest in different HIV-specific PROMs [27,30,32]. The survey for HCPs (55 items) (Appendix A) was shorter. It documented their demographic characteristics and addressed our first objective, by collecting information on their perspective on smart device use, healthcare applications, and self-efficacy [33]; our second objective, by capturing interest in Opal’s functions [6,27,32]; our third objective, by assessing the anticipated impact of PLWH access to physicians’ clinical notes [31] and the anticipated compatibility of Opal with their work [34]; and our fourth objective, by measuring interest in different types of HIV-specific PROMs [27,30,32].

Item response options included multiple choice and 5 to 7-point Likert scales. For items with 5 or 6-point Likert scales, responses were collapsed into three categories: “not at all interested” and “not interested” were combined as “not interested”; neutral responses “I don’t know” and/or “undecided” (6-point Likert scales included both responses) were classified as “undecided”; while positive responses “a little interested” and “very interested” were classified as “interested”. For items with a 7-point Likert scale, responses were collapsed into three categories: negative responses (“completely disagree”, “disagree”, and “somewhat disagree”) were classified as “disagree”; the neutral response (“undecided”) remained as “undecided”; and positive responses (“agree”, “somewhat agree”, and “completely agree”) were classified as “agree”. 

Participants were provided an in-person 5-minute PowerPoint presentation on Opal’s main functions (Appendix A) and were offered a chance to ask questions to ensure participants fully understood how Opal may fit into their care or work. HCPs were also introduced to how Opal could be used in their work to support their management of PLWH, for example, through using a clinic check-in system, or integrating data from PROMs into the clinical encounter. HCPs had also participated in focus group discussions prior to completing their surveys; however, results from the focus groups will be presented separately. A researcher administered the in-person survey to PLWH electronically, by presenting PLWH with each item and their possible responses before recording each answer, while HCPs completed a paper survey. Data were then entered into a secure online platform, REDCap© (version 9.1.15, Vanderbilt University, Nashville, TX, USA), which facilitated ease of administration and storage of data [35,36]. 

### 2.5. Statistical Analysis

Descriptive analysis was performed using R statistical software (version 1.2, R Foundation for Statistical Computing, Vienna, WIEN, Austria) [37]. The distribution of continuous variables was described by their means, standard deviations, and ranges; for categorical variables, relative frequencies were reported. To express uncertainty in estimates of proportions, 95% confidence intervals were reported.

## 3. Results

### 3.1. Sample Characteristics

Table 1 shows the characteristics of the PLWH and HCP participants. PLWH (*n* = 114) included 86 men (74%), 28 women (24%), and 2% identified as transgender or “other”. A total of 106 (93%) PLWH were recruited from the CVIS (Montreal), while 8 (7%) were from SMIT (Paris). CVIS clinic population data from 2019 shows that of the 1679 registered PLWH, 63% were men and 37% were women. Their mean age was 51.2 years old (SD = 12.7), compared with 47.8 (SD = 12.4) in the present study sample.

HCPs’ (*n* = 31) mean age was 46.5 years old (SD = 11.4). They included 20 women (65%) and 10 men (32%). Of the HCPs recruited, 16 (52%) were from the CVIS (Montreal), 8 (26%) were from non-CVIS Montreal sites, and 7 (22%) from SMIT (Paris).

### 3.2. Smart Device Ownership and Experience and Comfort Using Healthcare Applications

Overall, 96% of PLWH and 100% of HCPs owned at least one type of smart device, including computers (desktops or laptops), smartphones, and/or tablets. These three devices are capable of operating Opal, although the patient-facing side of Opal can be operated through smartphones and tablets only. PLWH still demonstrated high ownership (93%) when accounting for only these two devices; however, smartphone and tablet use for PLWH above 50 years old was lower (85%).

There were 82% of PLWH and 61% of HCPs who indicated very little to no experience using healthcare applications, including any applications targeted towards improving user health (for example, other patient portals, calorie counters, step counters, etc.); however, 74% of PLWH were willing to use Opal, and 68% of HCPs were willing to use Opal in their work to support the management of PLWH. For HCPs, this would entail using Opal to facilitate HIV care. Of those willing to use Opal, 80% of PLWH and 60% of HCPs reported very little to no experience with healthcare applications. With Opal, 61% of PLWH wanted immediate and comprehensive access to their medical records, while 25% preferred to only access information after review with their HCPs (see Table 2).

Most participants reported the capacity to use healthcare applications (90% PLWH, 90% HCPs), indicating PLWH’s ability to access smart devices and operate their healthcare applications. Of the 10 PLWH who did not agree they could use healthcare applications, three were over the age of 50 years. Among the three HCPs who did not feel capable of using healthcare applications, two were over 50 years old. 

Additionally, the proposed Opal patient portal was perceived as appealing by most participants (90% PLWH, 97% HCPs), and was met with approval by 89% of PLWH and 87% of HCPs.

### 3.3. Interest in Opal Functions and Preferences for Sharing Personal Health Information

The Opal functions that most interested the two groups included the appointment schedule (94% PLWH, 97% HCPs), user account and password (92% PLWH, 74% HCPs), and notifications and reminders (92% PLWH, 87% HCPs) (see Figure 1). Among the functions deemed more useful by HCPs than PLWH were a navigational tool (63% PLWH, 87% HCPs) and text messaging (62% PLWH, 77% HCPs). Compared to PLWH, HCPs were less interested in functions for PLWH to access treatment plans (89% PLWH, 64% HCPs), access consultation notes (85% PLWH, 39% HCPs), and share consultation notes (85% PLWH, 52% HCPs). 

Using the Opal patient portal, PLWH would have the option to share their personal health information. PLWH were most comfortable sharing their HIV health data with their primary HIV healthcare provider (96%, 95% CI = 90, 99), followed by pharmacists (75%, 95% CI = 66,83) and other HIV specialists at their clinic (75%, 95% CI = 66, 83) (see Figure 2). However, PLWH were more reluctant to share information with public health (45%, 95% CI = 35, 54) and health insurers (36%, 95% CI = 27, 45). 

### 3.4. Anticipated Benefits and Inconveniences of Opal

Most PLWH believed Opal could provide various benefits, including better preparing themselves for clinical visits (89%, 95% CI = 82, 94), remembering their HIV care plan (87%, 95% CI = 79, 92), and feeling more in control of their healthcare (87%, 95% CI = 79, 92) (see Figure 3). However, nearly one-third of PLWH (36%, 95% CI = 27,45) had no concerns about their privacy if using the Opal patient portal. 

As for HCPs, almost two-thirds of physicians (62%) were worried PLWH would contact them with questions about consultation notes, nearly half (46%) of physicians had concerns PLWH may find significant errors in their consultation notes, and 46% of physicians were concerned PLWH would request changes to their consultation notes. Lastly, approximately two-thirds (64%) of all HCPs thought Opal would fit into the way they work. 

### 3.5. Interest in Different Patient-Reported Outcome Measures

At least 60% of all participants were interested in each of the PROM types evaluated (see Figure 4), particularly those regarding the experience of healthcare (96% PLWH, 97% HCPs), HIV self-management (92% PLWH, 97% HCPs), and the experience of treatment (90% PLWH, 90% HCPs). The PROM types of least interest were body and facial appearance (68% PLWH, 62% HCPs) and disability (62% PLWH, 81% HCPs). 

## 4. Discussion

This study sought to ascertain the feasibility of using the Opal patient portal in HIV care with key stakeholder input obtained through a cross-sectional survey. These results highlight a high prevalence of smart device ownership, interest in using Opal, acceptability of most Opal portal functions, several perceived benefits and inconveniences of Opal, and acceptability of most PROMs. Considering these results, Opal may be feasible for use in HIV care. 

### 4.1. Smart Device Ownership, Experience with Healthcare Applications, and Health Info Preferences

Critical to implementing a patient portal is the consideration of factors such as user access to smart devices, experience with healthcare applications, and willingness to use a patient portal [38,39,40,41,42]. Our sample revealed a high use of smart devices, through which participants could access Opal, across all age groups for all participants. However, the uptake of and access to smartphones or tablets was relatively lower in age groups above 50 years (85% of PLWH) compared to younger age groups, which corroborates prior studies noting lower access to and uptake of patient portals with older age [40,41]. The mean age of the entire CVIS clinic is also above 50 years old and may affect the overall uptake of Opal. Additionally, most participants had limited healthcare application experience, although this did not reduce participants’ interest in using the patient portal. Among PLWH, this interest is encouraging; however, given participants’ limited healthcare application experience, clinician leadership and promotion of Opal are important considerations in facilitating its uptake [12]. 

Interestingly, the proportion of PLWH preferring immediate access to medical records and Opal access after physician review were similar to the preferences of oncology patients in the initial Opal study [6]. It is important to consider Opal’s initial success in oncology care, which utilized a patient-centered approach where patients could choose their preferred level of access to personal health information. Given the varying preferences for access to personal health information in the HIV care context, it would be imperative to offer PLWH the option to choose their preferred level of access to personal health information during end-user testing to optimize PLWH uptake and satisfaction with Opal. Overall, most participants perceived Opal to be appealing. However, there was a difference between PLWH who would welcome Opal (89%), versus PLWH who were willing to use the patient portal (74%). This may be explained by PLWH who commented that although they may not use Opal themselves, other PLWH may benefit from using such a portal, and thus, they would still welcome Opal. 

### 4.2. Interest in Opal’s Functions

PLWH were interested in most proposed Opal functions including access to their treatment plan, consultation notes, and sharing consultation notes; however, HCPs were less receptive to these functions. These concerns mirror those reported in prior literature; specifically, HCPs worry that their workload will increase due to an influx of PLWH messages or phone calls with these types of portal functions [43]. However, other studies that examined actual patient portal usage showed that allowing access to consultation notes through patient portals does not increase clinician workload, and in some cases, it even decreases the need for telephone calls and may reduce unnecessary appointments [44,45].

### 4.3. Anticipated Benefits and Inconveniences of Opal

Patient portals are reported to be useful for monitoring the health of PLWH, as they could meet the changing needs and expectations of PLWH [46,47]. The anticipated benefits of Opal for PLWH, such as allowing for better clinic visit preparation and understanding of their HIV diagnosis, are consistent with these observations [46]. Despite the many anticipated benefits of using Opal, there were concerns as well. Opal raised privacy issues for many PLHW surveyed, which is congruent with prior literature citing concerns with data security theft, confidentiality, privacy, and HIV-related stigma as barriers to patient portal use and implementation [12,43,47,48,49,50,51,52]. 

### 4.4. Interest in Different Types of PROMs

Participants’ interest in types of PROMs [30] demonstrates the various topics and issues they prefer to discuss, particularly PLWH-perceived experiences of healthcare, symptoms, psychological challenges, and social support. PROMs capturing body and facial appearance received the least amount of interest amongst all participants, as some PLWH have not been exposed to the complications of outdated ART regimens. Additionally, PROMs capturing disability received less interest from PLWH compared to HCPs. Interestingly, compared to HCPs, PLWH showed less interest in PROMs related to psychological challenges and resources, as well as to HIV-related stigma, despite their well-documented prevalence among PLWH [53]. 

### 4.5. Limitations

A limitation of this feasibility study is the oversampling of male participants. Nevertheless, this is congruent with the predominantly male (63%) population of the CVIS clinic population, where most participants were recruited. In addition, there was a lack of equivalent participant recruitment from all sites to allow for site-to-site comparison; however, our goal was to implement Opal at the CVIS first, therefore recruitment was predominantly from the CVIS. Additionally, our use of convenience sampling of PLWH may result in volunteer bias and social desirability bias, while purposeful sampling of HCPs may have led to a sampling bias. However, our sampling of HCPs was intended to include a variety of HIV-related healthcare specialists in terms of role and expertise. 

Lastly, the participant sample from France is limited. Still, we aim to utilize this data to inform an upcoming Opal pilot in France. Moreover, our team conducted concurrent focus group discussions in Montreal and France that revealed congruent results. For these two reasons, we decided to include the data from the two focus groups conducted in France.

### 4.6. Future Considerations

Understanding the feasibility of using a patient portal and the needs of PLWH and their HCPs was an initial step prior to piloting the implementation of Opal in HIV clinical care. We will continue engaging with key stakeholders to optimize the Opal patient portal for pilot development and testing. To optimize Opal for use in HIV care, we will discuss further design considerations with stakeholders that could optimize portal uptake, utility, and usability. 

## 5. Conclusions

This study assessing the feasibility of adapting Opal to HIV care revealed several considerations for using a patient portal for PLWH and their HCPs, primarily for a large HIV clinic such as the CVIS. The results obtained suggest that Opal’s implementation at the CVIS is feasible, considering the high rate of smart device ownership, comfort with using healthcare applications, anticipated benefits of using Opal, and interest in most Opal functions and proposed PROMs. Opal may personalize HIV care by incorporating PROMs and functions that are important to PLWH, while maintaining a secure and confidential platform. By consulting key stakeholders, who will eventually be end-users of Opal, this study may also offer insight into a framework for future patient portal adaptations from one specialty to another.

## Figures and Tables

**Figure 1 jpm-11-00134-f001:**
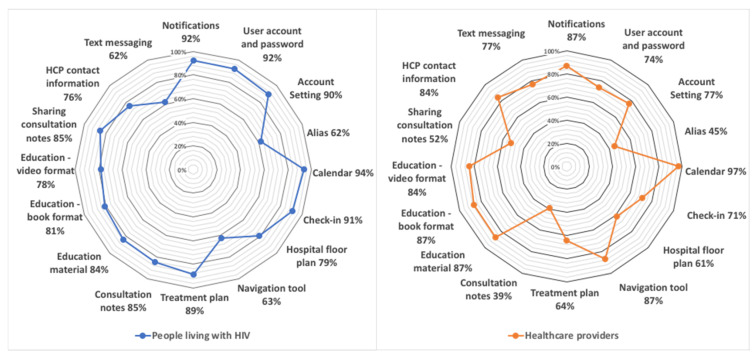
Participant interest in proposed Opal functions.

**Figure 2 jpm-11-00134-f002:**
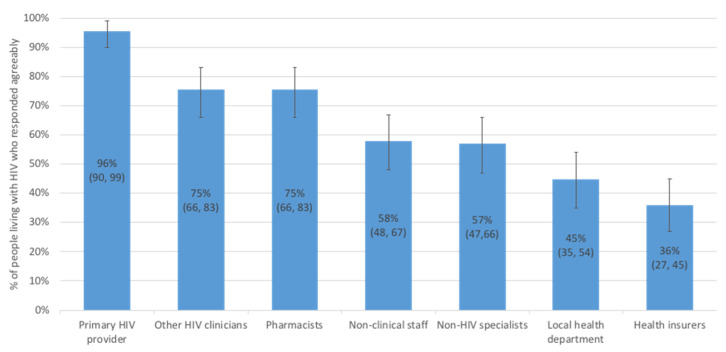
Preferences of people living with HIV for sharing their personal health information with others. Percentages are displayed with 95% confidence intervals in brackets.

**Figure 3 jpm-11-00134-f003:**
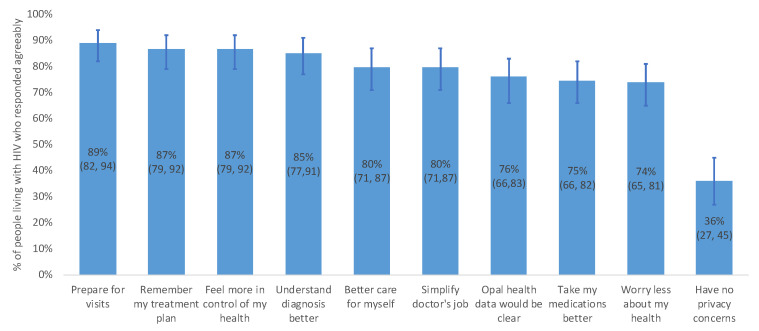
People living with HIV who responded agreeably to anticipated benefits and inconveniences of Opal. Percentages are displayed with 95% confidence intervals in brackets.

**Figure 4 jpm-11-00134-f004:**
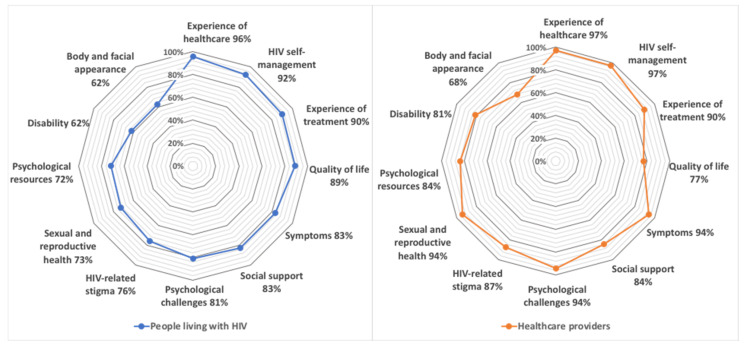
Participant interest in types of HIV-specific PROMs for administration via Opal.

**Table 1 jpm-11-00134-t001:** Descriptive characteristics of people living with HIV and healthcare providers who participated in the study.

	People Living with HIV (*n* = 114) Mean (SD) or %	Healthcare Providers (*n* = 31) Mean (SD) or %
Age (years)	47.8 (12.4)	46.5 (11.4)
Range	27, 74	25, 68
Gender		
Male	74	32
Female	24	65
Other/Transgender	2	3
Sexual orientation		
Heterosexual	48	N/A
Men who have sex with men	41	
Bisexual	9	
Unsure or other	2	
Marital status		
Single	47	N/A
Married	38	
Divorced or widow(er)	15	
Level of education		
University or higher	36	N/A
CEGEP *, trade/vocational school, or high school	53	
Up to high school	11	
Paid work		
Student	8	N/A
Part-time	14	
Full-time	35	
Unemployed, retired, or disabled	43	
Income (CAD)		
1 to 19 999	33	N/A
20 000 to 39 999	29	
40 000 to 59 999	15	
>60 000	14	
None or missing	9	
Ethnicity		
Caucasian or White	36	N/A
Black, African, or Carribean	31	
Latino, Latin American, or South American	20	
North African or Middle Eastern	6	
Asian or Pacific Islander	<5	
Indian or South Asian	<5	
Other	<5	
Aboriginal, First Nations, or Métis	---	
Occupation		
Physician	N/A	42
Pharmacist		26
Nurse		19
Social worker		6
Administrative staff		6
Smart devices owned		
Smartphone	90	87
Computer (desktop or laptop)	65	84
Tablet	39	29
iPod or phablet	8	19
Smartwatch	<5	10
Other	<5	<5
None	<5	---

N/A represents not applicable, --- represents no responses. * CEGEP is the first level of post-secondary education exclusive to Quebec, Canada.

**Table 2 jpm-11-00134-t002:** Participants’ healthcare application experience, willingness to use a patient portal, preferences for accessing medical records, healthcare application self-efficacy, and acceptability of Opal.

	People living with HIV (*n* = 114) %	Healthcare providers (*n* = 31) %
Healthcare application experience		
None to very little	82	61
Moderate to extensive	18	35
Willing to use a patient portal		
Yes	74	68
No	22	10
Uncertain	4	19
Access to medical records		
Immediate access	61	N/A
Following physician review	25	
No access	10	
Only need-to-know information	4	
Healthcare application self-efficacy		
Capacity to use healthcare applications	90	90
Comfortable using healthcare applications	77	81
Ease of healthcare application use	76	77
Confidence pressing the right buttons to promote health	73	84
Acceptability of the proposed Opal patient portal		
Opal is appealing	90	97
Opal has my approval	89	87
I would welcome Opal in HIV care	89	81
I like Opal	76	81

N/A represents not applicable.

## Data Availability

Data sharing is not applicable to this article.

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
