# Peer review of "Acceptability of a Patient Portal (Opal) in HIV Clinical Care: A Feasibility Study"

_jpm, 2021, doi:10.3390/jpm11020134_

Round 1

Reviewer 1 Report

The study by Chu et al., with the title " Acceptability of a patient portal (Opal) in HIV clinical care: a feasibility study", aims to investigate the feasibility of the use of a patient portal (OPAL) in HIV care in Montreal, Canada and Paris, France. Most importantly the primary outcome of the study is to assess and evaluate acceptability of OPAL by PLWH and HCP. The study is sound as patient portal could increase adherence to ART and provide additional tools for PLWH to easily access their medical records and assess the quality of care they receive. Moreover OPAL seems to be an important tool for HIV personalized medicine.

Comments:

  1. This study was not a multicenter study because the number of participants in France was too small to evaluate OPAL acceptability in France. Furthermore, the REB was not applied for and approved in France. Therefore, the author should consider this study to be only focused on Montreal and Canada.
  2. Please introduce the acronym PLWH and HCP on first use in the abstract and in the body text.
  3. What makes OPAL innovative and different from existing Patient Portals all across Canada?
  4. Please revise the introduction (page 2, lines 85-91): The primary objective of the study is to evaluate ................(Montreal, Quebec). The study aims to assess:
  • The experience of PLWH and HCP with healthcare application and smart device ownership.
  • PLWH and HCP interest in Opal and willingness to share personal information
  • The anticipated Opal benefits and inconveniences
  • PLWH and HCP interest in different Opal PROMS.
  1. The author could also at the end of the introduction clearly state the primary outcome of the study (acceptability) and the anticipated research question.
  2. Study design and sites: The author should consider dropping the France data as the number of participants was to small and the REB not obtained. The small sample of France might have skewed the Canadian data (update your statistics accordingly).
  3. The REB was approved at McGill University Health Centre. How about REP approval in France?
  4. Please change "classed" to "classified" or "labelled"; page 3, line 144.
  5. Is 5 min presentation enough for participants to understand the App?
  6. Results: Why is there an oversampling of male.
  7. The Paris sample size (8 participants, 7%) is problematic and put into question the feasibility and acceptability of the OPAL portal in France. As suggested it would be best to evaluate a separate feasibility in France with more PLWH and HCPs.
  8. Table 2: How does the author explains the fact that 22% of PLWH who participated in the study are not willing to use a patient portal but 89% will welcome OPAL in HIV care.
  9. Is the author suggesting that age was a factor in the ability to use healthcare application? In the context of this paper only 3 and 2 PLWH and HCP respectively did not agree to use healthcare application.
  10. Please write confidence interval (CI) in the result section as well and not only in the figure (figure 2 and 3).
  11. Please restructure the following sentence: "while .............compare to HCPs", page 9, lines 303-304.

Author Response

Hello, please see the attachment for comments below. 

Reviewer 2 Report

Overall comments

The present study surveys the disposition of people living with HIV (PLWH) and their healthcare practitioners (HCPs) towards using a personalized smartphone app (Opal) for disease management. The study design is clear and the manuscript is well written with a good body of supporting literature. The authors compare the responses between PLWH and HCPs and discuss interesting areas of overlap and divergence with figures that neatly summarize and explore the data. I have only minor comments, mostly to improve the clarity of certain areas (in particular, the tables and figures). I thank the authors for the opportunity to review their work; I hope my comments may be of use and that the app will be a great success in supporting PLWH and their HCPs.

Improvements

  1. Abstract. Although defined later, please define abbreviations “HIV”, “PLWH” and “HCPs” here.
  2. Introduction, line 55. The authors state “To support PLWH and their HCPs in the management of HIV, a promising solution is …”. ‘Support’ is mentioned, but I could not clearly parse the problem that the “promising solution” was addressing in this sentence. Do the authors mean “To better support PLWH and their HCPs in the management of HIV, we developed a patient portal …”, or “A patient portal with [x, y and z] features may address these problems faced by PLWH and their HCPs; thus, we developed …”?
  3. Methods, line 98. The authors state “…which coincide with our objectives”. This could make it sound as if the study was conceived without a clear connection between methods and objectives. Possibly better to say simply “… which were among our objectives” or similar.
  4. Methods, line 159. “… which conferred…” may be better as “enabling” or “facilitating”.
  5. Table 1, income section. The “none” in “None or missing” appears to overlap with “<20 000”, i.e., $0. Were these values counted twice? Please revise.
  6. Table 1, column 3, second last row. Dollar sign ($) is shown, should this be “4”?
  7. Table 1, last three rows. Values are listed as “<5”, but I am puzzled as to why the actual values are not stated explicitly (like every other value in the table).
  8. Table 1. Is there a reason why the possibly important data collected in survey question “What ethnic group(s) or family background(s) do you identify with?” (Supplemental File 1) is not represented in Table 1 or discussed?
  9. Figures 2 and 3. Please label the y-axis. Please also specify what the error bars and bracketed numbers refer to. It appears that the authors have attempted to do the latter in the line following the figure caption, but this looks like a separate paragraph.
  10. Lines 226 and 234. Furthermore, is confidence interval the appropriate statistical metric to report for this data? One standard deviation or range seem more appropriate here.
  11. Figure 2, Figure 3 captions. Please consistently use “PLWH” or “people living with HIV”.

Author Response

Hello,

Please see the attached comments below. 

Thank you so much.
